Therapeutic effect of and mechanisms underlying the effect of miR-195-5p on subarachnoid hemorrhage-induced vasospasm and brain injury in rats

Tsai Tai-Hsin 1 2 3
Chang Chih-Hui 1 2 3
Lin Szu-Huai 4
Su Yu-Feng 1 2 3
Tsai Yi-Cheng 1 3
Yang Sheau-Fang 5 6
Lin Chih-Lung chihlung1@yahoo.com 1 2 3
1 Division of Neurosurgery, Department of Surgery, Kaohsiung Medical University Hospital , Kaohsiung , Taiwan
2 Department of Surgery, School of Medicine, College of Medicine, Kaohsiung Medical University , Kaohsiung , Taiwan
3 Graduate Institute of Medicine, College of Medicine, Kaohsiung Medical University , Kaohsiung , Taiwan
4 Department of Nursing, Kaohsiung Medical University Hospital , Kaohsiung , Taiwan
5 Department of Pathology, Kaohsiung Medical University Hospital , Kaohsiung , Taiwan
6 Department of Pathology, School of Medicine, College of Medicine, Kaohsiung Medical University , Kaohsiung , Taiwan
Piato Angelo
Electronic publication date: 2021 Jun 22
Publication date: 2021
Volume: 9
Electronic Location ID: e11395
Received 2020 Dec 3; Accepted 2021 Apr 12
Copyright: ©2021 Tsai et al.
Copyright year: 2021
Copyright holder: Tsai et al.
License: This is an open access article distributed under the terms of the Creative Commons Attribution License, which permits unrestricted use, distribution, reproduction and adaptation in any medium and for any purpose provided that it is properly attributed. For attribution, the original author(s), title, publication source (PeerJ) and either DOI or URL of the article must be cited.
License URL: https://creativecommons.org/licenses/by/4.0/

Keywords: Subarachnoid hemorrhage, Cerebral vasospasm, Apoptosis, iNOS, eNOS, miR-195-5p, TNF-α, NF-κB

Funding: National Science Council, Taiwan 101-2314-B-037-018-MY2 Kaohsiung Medical University Hospital KMUH 101-1R16 KMUH 102-2R17 This work was supported by the National Science Council, Taiwan, under Grant NSC 101-2314-B-037-018-MY2, and Kaohsiung Medical University Hospital (KMUH 101-1R16 and KMUH 102-2R17). The funders had no role in study design, data collection and analysis, decision to publish, or preparation of the manuscript.

==============================
Objectives

There is much evidence suggesting that inflammation contributes majorly to subarachnoid hemorrhage (SAH)-induced cerebral vasospasm and brain injury. miRNAs have been found to modulate inflammation in several neurological disorders. This study investigated the effect of miR-195-5p on SAH-induced vasospasm and early brain injury in experimental rats.

Methods

Ninety-six Sprague-Dawley male rats were randomly and evenly divided into a control group (no SAH, sham surgery), a SAH only group, a SAH + NC-mimic group, and a SAH + miR-195-5p group. SAH was induced using a single injection of blood into the cisterna magna. Suspensions containing NC-mimic and miR-195-5p were intravenously injected into rat tail 30 mins after SAH was induced. We determined degree of vasospasm by averaging areas of cross-sections the basilar artery 24h after SAH. We measured basilar artery endothelial nitric oxide synthase (eNOS), inducible nitric oxide synthase (iNOS), nuclear factor kappa-light-chain-enhancer of activated B cells (NF-κ B), phosphorylated NF-κ B (p-NF-κ B), inhibitor of NF-κ B (Iκ Bα) and phosphorylated-Iκ Bα (p-Iκ Bα). Cell death assay was used to quantify the DNA fragmentation, an indicator of apoptotic cell death, in the cortex, hippocampus, and dentate gyrus. Tumor necrosis factor alpha (TNF-α) levels were measured using sample protein obtained from the cerebral cortex, hippocampus and dentate gyrus.

Results

Prior to fixation by perfusion, there were no significant physiological differences among the control and treatment groups. SAH successfully induced vasospasm and early brain injury. MiR-195-5p attenuated vasospasam-induced changes in morphology, reversed SAH-induced elevation of iNOS, p-NF-κ B, NF-κ B, and p-Iκ Bα and reversed SAH-induced suppression of eNOS in the basilar artery. Cell death assay revealed that MiR-195-5p significantly decreased SAH-induced DNA fragmentation (apoptosis) and restored TNF-α level in the dentate gyrus.

Conclusion

In conclusion, MiRNA-195-5p attenuated SAH-induced vasospasm by up-regulating eNOS, down-regulating iNOS and inhibiting the NF-κ B signaling pathway. It also protected neurons by decreasing SAH-induced apoptosis-related cytokine TNF-α expression in the dentate gyrus. Further study is needed to elucidate the detail mechanism underlying miR-195-5p effect on SAH-induced vasospasm and cerebral injury. We believe that MiR-195-5p can potentially be used to manage SAH-induced cerebral vasospasm and brain injury.

Background

Although aneurysmal subarachnoid hemorrhage (SAH) induction of cerebral vasospasm has been recognized for more than half a century and many important advances in surgical technique and intensive neurological critical care have been made over that time, cerebral vasospasm remains the leading cause of mortality and morbidity in patients after aneurysmal SAH. Approximately one half those who survive SAH remain permanently disabled by cognitive dysfunction and do not return to work (Kassell et al., 1985; Longstreth Jr. et al., 1993). Vasospasm alone cannot explain all the subtle changes in behavior and memory in SAH patients. There may other explanations, including global ischemia-induced apoptosis (Cahill, Calvert & Zhang, 2006), which has been found to occur in the hippocampus, blood–brain barrier and vasculature with varying degrees of necrosis (Ostrowski, Colohan & Zhang, 2005; Park et al., 2004) and has been implicated in vasospasm and smooth muscle cell proliferation in spastic arteries (Zubkov et al., 2000). It also been implicated in aneurysmal formation and rupture in animal models and humans (Kassell et al., 1985). Thus, cell death may have important consequences on both vasospasm and long-term sequelae following SAH.

The mechanisms underlying SAH-induced vasospasm are not fully understood and no adequate treatment of this vasospasm has been found. SAH has been found to impair endothelium dependent relaxation in large cerebral arteries in various animal models (Yang et al., 2001) and in the basilar arteries of SAH patients (Sayama, Suzuki & Fukui, 1999). After SAH, the production or activity of NOS is has been to found to be altered probably, a consequence of reductions in eNOS and increases in iNOS in vasospastic cerebral vessels (Yang et al., 2001). Khurana et al. (2004) have found an association between basilar artery recombinant eNOS levels and relief of SAH-induced vasospasm. Estrogen has been found able to mediate vasodilation and neuroprotection in experimental SAH rats by activating eNOS and preventing SAH-induced increases in iNOS, which suggests that eNOS and iNOS may play different roles in mediating vascular tone after SAH (Lin et al., 2009). One review has concluded that vascular spasm alone cannot explain all the SAH-related morbidity and suggests that inflammation is becoming a promising therapeutic target for studies investigating possible strategies for treating vasospasm and brain injury following SAH (Provencio, 2013). Early after SAH is induced experimentally, there are increases in vascular endothelial growth factor (VEGF), mitogen-activation protein kinase (MAPK) and matrix-metalloproteinase-9 (MMP-9) (Kusaka et al., 2004; McGirt et al., 2002). It is interesting to note that depletion of myeloid cells prior to SAH protects against vasospasm and behavioral deficits caused by SAH in mice (Provencio et al., 2011).

MiR-195-5p, a member of the miR-15 family and a known tumor suppressor, has been found to be downregulated in several cancers, including prostate, lung, colon, breast, and oral squamous cell carcinoma (Giordano et al., 2019). Much is known about miR-195-5p. For example, it can potentially be used as biomarker for membranous nephropathy (Zhou et al., 2019), it can regulate dermal papilla cell hair follicle inductivity (Zhu et al., 2018), it can promote pulmonary arterial smooth muscle cell proliferation and migration (Zeng et al., 2018), it has been found to be upregulated in patients with deep vein thrombosis and be involved in regulating apoptosis of vascular endothelial cells (Jin et al., 2019), and it has been found to inhibit angiogenesis in preeclampsia (Sandrim et al. 2016). Considered together, these findings highlight the role miR-195-5p plays in regulating both cell growth and apoptosis. Recently, the elevated expression of miRNA-195-5p has been associated with decreases in hypoxia inducible factor 1alpha (HIF-1 α) levels and increases in vascular endothelial growth factor (VEGF) serum levels (Giordano et al., 2019). Moreover, it has been suggested that circulating miR-195-5p might be used as a biomarker and therapeutic target for transient ischemic attack and acute ischemic stroke (Giordano et al., 2019).

MiR-195-5p also has anti-inflammatory and anti-apoptosis effects in addition to its ability to regulate cell growth (Bras et al., 2017; Chang et al., 2020; Cheng et al., 2019; Yang, Cao & Zhang, 2020). MiR-195-5p exerts its anti-inflammatory effects by blocking the NF-κB pathway (Bras et al., 2017) and inhibiting the production of pro-inflammatory cytokines in macrophages (Bras et al., 2017). Furthermore, it inhibits C-X3-C motif chemokine receptor 1(CX3CR1)-mediated neuroinflammation making possible improved nerve cell survival in the context of chronic cerebral ischemic damage (Yang et al., 2018). MiR-195-5p can also inhibit apoptosis in damaged neural cells and promote the neural stem cells for neurogenesis (Cheng et al., 2019). It has been found to attenuate neuronal apoptosis in ischemic stroke rats (Chang et al., 2020) and protect against ischemic stroke clinically (Bras et al., 2017).

Although well-studied, the possible role of this MiRNA in the treatment of SAH-induced vasospasm and brain injury has never been explored. MiR-195-5p could possibly be used to attenuate SAH-induced vasospasm, apoptosis and secondary brain injury because it has neuroprotective, anti-inflammatory and anti-apoptosis effects. In this study, the first of its kind, we investigate the possible therapeutic effect that miR-195-5p might have on SAH-induced vasospasm and apoptosis in experimental SAH rats and the mechanisms through which it might exert its effect.

Material and Methods

Animals

The procedures for treatment of the animals used in this study are approved by the Kaohsiung Medical University Animal Care and Use Committee (IACUC approval No: 100201). The experimental procedures are based on the previous article (Lin et al., 2003; Tsai et al., 2020). Ninety-six male Sprague-Dawley rats (330 to 370 g), were purchased from BioLASCO Taiwan Co., Ltd. They were housed individually in a temperature (25 °C) and humidity (40–50%) controlled rooms with a 12 h light/dark cycle and provided unlimited access to food and water. To induce SAH, we first administered appropriate anaesthesia and analgesia to the animal to minimize pain. The animals were anesthetized with an intraperitoneal injection of 40 mg/kg Zoletil 50® containing a mixture of zolazepam and tiletamine hypochloride (Virbac, Carros, France). Rectal temperature was controlled at 37 ± 0.5 °C with a heating pad (Harvard Apparatus). The tail artery was cannulated with a polyethylene catheter so that we could monitor blood pressure and heart rate. Twenty-four hours after this induction of SAH, some brain tissues were removed after normal saline perfusion. Animals were kept warm using heating pad and using bedding during post-surgical recovery. They were checked frequently, about every 10–15 min, and were turned side to side until they recovered. The animals were then profoundly anaesthetized before they were euthanized by perfusion and fixation. All rats were euthanized humanely with intraperitoneal injection of Zoletil 50®. Basilar artery and brain tissue samples were collected after intra-cardiac perfusion with normal saline and the brain was removed, placed in a fixative solution, and stored at 4 °C overnight.

Induction of experimental SAH

This study adopted a single-hemorrhage rodent model of SAH. The rod model of SAH was described by Lin et al. in a previous journal (Lin et al., 2003; Tsai et al., 2020). Niety-six rats were randomly and evenly divided into four groups: a control group (no SAH, with sham surgery) (n = 24), a SAH only group (n = 24), a SAH + NC-mimic group (n = 24), and a SAH + miR-195-5p group (n = 24). Blood was injected to blood into the cisterna magna to induce SAH and 30 min suspensions containing NC-mimic and miR-195-5p (in vivo-jetPEI®; Polyplus Transfection Co.) were intravenously injected into the tails of the rats to create the latter two groups.The cisterna magna was punctured percutaneously with a 25-gauge butterfly needle. About 0.10 to 0.15 ml of cerebrospinal fluid was slowly withdrawn and the junction between the needle and tube was clamped. Fresh autologous, nonheparinized blood (0.3 ml) was withdrawn from the tail artery. Using a needle-in-needle method (inserting a 30-gauge needle into the 25-gauge butterfly needle at the junction of needle and tube), we slowly injected blood into the cisterna magna. Basilar artery and brain tissue samples were collected after intra-cardiac perfusion with normal saline for the assessment of protein expression of iNOS, eNOS, phospho-NF-κB (p-NF-κB), NF-κB and phospho-IκB α (p-IκB α) in basilar arteries, the performance of cell death assay analysis, and the measurement TNF-α protein in the brain.

Basilar artery morphometric analysis

We used 24 animals from each group to perform a morphometric analysis of the basilar artery. Measurements were performed by an investigator blinded to the treatment groups. At least five random arterial cross-sections from each animal were evaluated qualitatively and the cross-sectional area of each section was measured using a computer-assisted image analysis system (ImageJ, NIH). The areas of the five cross-sections from basilar artery were averaged to provide a single value for each animal. The areas were calculated by measuring the perimeter of the actual vessel lumen and then calculating the area of an equivalent circle (area =∂r2, where r is the radius) based on the calculated equivalent r value from the perimeter measurement (r = perimeter/2 ∂), correcting for vessel deformation and off-transverse sections (Kida, Rothman & Hyder, 2007).

Cell death assay

We used 24 animals from each group for our cell death assay. A commercial enzyme immunoassay was used to identify cytoplasmic histone-associated DNA fragments (Cell Death Detection ELISAPLUS, Roche) as a measurement of DNA fragmentation and apoptotic cell death. Fresh brain tissue samples from the cortex, hippocampus (CA1, CA2, and CA3), and dentate gyrus were collected. Protein was extracted from the cytosolic fraction by using ProteoJET™ Cytoplasmic and Nuclear Protein Extraction Kit (Fermentas). A cytosolic volume containing 50g protein was used for the enzyme-linked immunosorbent assay (ELISA) following manufacturer’s directions.

Western blot analysis of iNOS, eNOS, p-NF-κB, NF-κB and p-I κBα

We used 24 animals from each group to collect tissue samples from the basilar artery for Western blot analyses. To do this, the basilar artery was homogenized in ice-cold T-PER Tissue Protein Extraction Reagent (Thermo) with a protease inhibitor (cOmplete™ ULTRA Tablets; Roche) and phospho-protease inhibitor (PhosSTOP, Roche), and centrifuged 15,000 rpm for 20min. The protein concentration was estimated following the Bio-Rad protein microassay procedures. Briefly, samples were heated for 5 min in boiling water. Equal amounts of protein were loaded in each land of SDS-PAGE. The gels were transferred onto polyvinylidene difluoride (PVDF; PerkinElmer) membrane by electroblotting for 120 min (400 mA). The membrane was blocked overnight at 4 °C with the Tween-Tris buffer saline solution (t-TBS; 20 mM Tris base, 0.44 mM NaCl, 0.1% Tween 20, pH 7.6) containing 5% nonfat dry milk. The blot was incubated with primary antibodies: 1:500 dilution of the antibody against mouse monoclonal iNOS (BD); 1:1000 dilution of the antibody against mouse monoclonal eNOS (BD); a 1:500 dilution of the antibody against rabbit polyclonal p-NF-κB (Cellsignaling); 1:1000 dilution of the antibody against rabbit polyclonal NF-κB (Proteintech); 1:500 dilution of the antibody against mouse monoclonal p-IκBα(Cellsignaling); and 1:80000 dilution of the antibody against β-actin (Sigma, St. Louis, MO). They were rinsed with t-TBS for 30 min and incubated with goat anti-mouse IgG or anti-rabbit IgG antibody conjugated to horseradish peroxidase (Jackson ImmunoResearch). β-actin levels served as an internal standard and to account for loading differences. Membranes were rinsed with t-TBS for 30 min, incubated with electrochemiluminescence reagent (Western Lightning ECL Pro; PerkinElmer) for 2 min according to the manufacturer’s directions. The intensity of each band was quantified by ImageJ software (NIH), and protein levels were expressed as the ratio of the values of the detected protein bands to those of β-actin bands.

Measurement of tumor neurosis factor-alpha (TNF-α)

We used 24 animals in each group to measure TNF-α in various brain tissue samples. ELISA was used to measure TNF-α in tissue sampled from cerebral cortex, hippocampus (CA1, CA2 and CA3) and dentate gyrus. This was performed using a commercial kit (Biosource) according to the manufacturer’s instructions. Spectrophotometric analysis was carried out using the biotechnology software set at 450 nm to determine TNF-α determination following assay guidelines. Absorbance of the colored product was measured at 450 nm.

Statistics

The data were expressed as mean ± standard error of the mean and illustrated using GraphPad Prism software (version 8.0; GraphPad, Inc., La Jolla, CA, USA). Differences between the experimental groups were analyzed by one-way ANOVA was utilized, followed by a Tukey’s post hoc test. (IBM SPSS Statistics, version 24.0; IBM Inc., Armonk, New York, USA). A p-value of <0.05 was considered significant.

Results

Basilar artery cross-sectional luminal area measurements

The cross-sectional area of basilar arteries was significantly reduced in SAH animals subjected (Control, n = 6; SAH, n = 6; SAH +mimic, n = 5; SAH +miR-195-5p, n = 6). Compared with areas measured for the control group (74,484.1 ± 4,089.9 µm2), the areas in the SAH only group (46,470.0 ± 4434.7 µm2) and SAH plus NC-mimic group (42792.4 ± 4975.3 µm2) were reduced by 37.6% (p < 0.001) and 42.5% (p < 0.01), respectively. The areas measured for miR-195-5p group (77,428.2 ± 3,771.67 µm2) differed significantly from those of the SAH only group (p < 0.01). No significant area difference found between the miR-195-5p group and the controls (Fig. 1).

Figure 1 Bar graphs showing the effects of miR-195-5p on cerebral vasospasm in cross-sectional areas.

The average luminal area (mean ± SEM) of basilar arteries is shown for each group of animals. The basilar artery luminal areas of the miR-95-5p treated group were markedly larger than those in the SAH-only group (** p < 0.01, significantly different from the control group; ** p < 0.01 compared with the SAH-only group.).

Protein expressions of iNOS, eNOS, p-NF-κB, NF-κB and p-IκBα in basilar artery

Protein expression of iNOS was increased significantly in the SAH only and the SAH plus NC-mimic groups, compared to controls (Control, n = 6; SAH, n = 5; SAH+mimic, n = 4; SAH+miR-195-5p, n = 5; p < 0.001, p <0.001 and p = 0.084, respectively). The MiR-195-5p had a significantly decreased protein expression of iNOS (p <0.001), compared with SAH only groups. They were found to have significantly higher iNOS than the controls (p < 0.05) (Fig. 2).

Figure 2 Western blotting for basilar artery iNOS and β-Actin.

Bar graphs showing miR-195-5p treatment reversing the increase in iNOS protein expression within the basilar artery following SAH (* p < 0.05, *** p < 0.001 significantly different from the control group; ###p < 0.001 compared with the SAH-only group.).

The SAH only and SAH plus NC-mimic groups had significantly lower protein expression eNOS in basilar artery tissues, compared to controls (Control, n = 6; SAH, n = 5; SAH+mimic, n = 4; SAH+miR-195-5p, n = 5; p = 0.001, p = 0.004 and p = 0.240, respectively). That suppression of eNOS was significantly reversed by MiR-195-5p (p < 0.001, compared with SAH only group). There was no significant difference in eNOS between the miR-195-5p and control groups (Fig. 3).

Figure 3 Western blotting for basilar artery eNOS and β-Actin.

Bar graphs showing miR-195-5p treatment inducing the increase in eNOS protein expression within the basilar artery following SAH (** p < 0.01, significantly different from the control group; ##p < 0.01 compared with the SAH only group.).

The percentage of p-NF-κB and NF-κB protein was found to be significantly increased in both the SAH only and SAH plus NC-mimic groups (Control, n = 6; SAH, n = 5; SAH+mimic, n = 5; SAH+miR-195-5p, n = 5; p = 0.002 and p = 0.001, compare to controls), but significantly decreased the miR-195-5p group (p = 0.001, compared with SAH only group). We found no significant difference of percentage of p-NF-κB and NF-κB protein between the miR-195-5p and control groups (Fig. 4).

Figure 4 Western blotting for basilar artery p-NF-κB/NF-κB and β-Actin.

Western blotting for basilar artery p-NF-κB/NF-κ B and β-Actin. The percentage of p-NF-κ B /NF-κB was increased significant in the SAH only and SAH plus NC-mimic group. On the contrary, the percentage of p-NF-κ B/NF-κB was decreased significantly after miR-195-5p treatment (** p < 0.01, significantly different from the control group; ##p < 0.01 compared with the SAH only group.).

The protein expression of p-IκB α in basilar artery was found to be significantly increased in the SAH only and SAH plus NC-mimic groups as well as the miR-195-5p groups, compared with controls (Control, n = 4; SAH, n = 4; SAH+mimic, n = 4; SAH+miR-195-5p, n = 4; p < 0.001, p <0.001 and p = 0.145, respectively). The miR-195-5p was found significantly lower protein expression of p-IκBα than the SAH only group (p = 0.024) (Fig. 5).

Figure 5 Western blotting for basilar artery p-IκBα and β-Actin.

Western blotting for basilar artery p-IκBα and β-Actin. The protein content of p-IκBα was significantly increased in the SAH only and SAH plus NC-mimic groups, compared to the controls. However, a significant decrease in the protein content of p-IκBα was observed after miR-195-5p treatment (* p < 0.05, ** p < 0.01 significantly different from the control group; ##p < 0.01 compared with the SAH-only group.).

The anti-apoptotic effect of miR-195-5p in the dentate gyrus after SAH

DNA fragmentation was analyzed using a commercial enzyme immunoassay (cell death assay) 24 h after SAH. The cell death assay revealed that DNA fragmentation was significantly increased in the dentate gyrus in SAH only, SAH plus NC-mimic group and SAH plus miR-195-5p group (Control, n = 6; SAH, n = 5; SAH+mimic, n = 5; SAH+miR-195-5p, n = 5; p < 0.001, p  < 0.001 and p = 0.012, compared with controls). In the miR-195-5p treated group, it was significantly decreased compared with SAH only (p = 0.017). We found a significant difference in cell death between the miR-195-5p treat and control groups (p = 0.018), suggesting that while apoptotic cell death was increased in the dentate gyrus after SAH and miR-195-5p treatment had an anti-apoptotic effect in the dentate gyrus, the effect of the treatment was not complete. We found no significant change in DNA fragmentation in either the cortex or hippocampus between controls and the miR-195-5p-treated and the other SAH groups (Fig. 6).

Figure 6 Detection of apoptotic cell death after SAH.

(A) Detection of apoptotic cell death after SAH. The cell death assay revealed DNA fragmentation to be significantly increased in the dentate gyrus in SAH only and SAH plus NC-mimic groups. There was significantly less DNA fragmentation in the SAH plus miR-195-5p group that in SAH only group. However, there was no significant change in DNA fragmentation in the cortex and hippocampus between control and SAH with or without miR-195-5p treatment groups (* p < 0.05, ** p < 0.01, significantly different from the control group; #p < 0.05 compared with the SAH only group.).

The prevention effect of miR-195-5p in the SAH-induced up-regulation in TNF-α protein after SAH

We also examined the protein level of TNF-α in the cortex, hippocampus and dentate gyrus 24 h after the first SAH, compared to controls (Fig. 7). ELISA results showed no marked difference in the level of TNF-α in cortex and hippocampus between controls and SAH animals. However, TNF-α expression in the dentate gyrus in the SAH only and SAH plus NC-mimic groups were significantly higher that it was in the controls (Control, n = 5; SAH, n = 5; SAH+mimic, n = 5; SAH+miR-195-5p, n = 5; p <0.001 and p =0.001and p = 0.942, respectively). MiR-195-5psignificantly attenuated the SAH-induced increases in TNF-α in the dentate gyrus, compared to the SAH only groups (p = 0.001 ).

Figure 7 ELISA assay for TNF-α.

The expression of TNF-α was significantly elevated in the dentate gyrus in the SAH only and SAH plus NC-mimic groups. MiR-195-5p treatment significantly attenuated the SAH-induced increases in TNF-α in the dentate gyrus. There was no significant difference between the levels of TNF-α in SAH groups treated or untreated with miR-195-5p and the control animals in the cortex and hippocampus (** p < 0.01 compared to control group. #p < 0.05 compared to SAH only group.).

Discussion

The present study shows that miR-195-5p treatment prevented SAH-induced cerebral vasospasm in basilar artery and SAH-induced apoptosis in the dentate gyrus in rats. The mechanisms underlying miR-195-5p effect on SAH-induced vasospasm involve its up-regulation of expression of eNOS and down-regulation of iNOS in basilar artery after SAH. It reduced SAH-induced cell death in dentate gyrus by reversing SAH-induced increases in p-NF-κB/NF-κB, p-IκBα and TNF-α. Together these findings provide evidence suggesting that miR-195-5p could potentially be used therapeutically to treat vasospasm and brain injury following SAH, an effect that it achieved in this study though the NF-κB signaling pathway.

Cerebral vasospasm that follows aneurysmal SAH has been considered a leading treatable cause of mortality and morbidity for recognized for more than half a century, its pathophysiologic mechanism remains elusive (Cook, 1995). In this study, miR-195-5p attenuated vasospasm-induced changes in morphology reversed SAH-induced elevation of iNOS and reversed SAH-induced suppression of eNOS in the basilar artery. We found SAH to be followed by decreased expression eNOS and increased expression of iNOS the basilar arteries of SAH only and SAH plus NC-mimic groups. Similarly, Osaka et al. showed in a rat single-hemorrhage model that eNOS was significantly phosphorylated in the basilar arteries at an early stage from 1 to 2 days after SAH, accompanied by up-regulation of iNOS (Osuka et al., 2009). Unlike eNOS, iNOS is induced in the brain vascular tissue of the SAH rat (Prunell, Mathiesen & Svendgaard, 2004) and can be increased by NF-κB activation through a sequence of events (Prunell et al., 2005). Treatment with aminoguanidine, a selective inhibitor of iNOS, has been found to relieve the SAH-induced cerebral vasospasm in rats (Sayama, Suzuki & Fukui, 1999). 17 β-estradiol has been found to potentiate vasodilation by activating eNOS, preventing increased iNOS activity caused by SAH, and decreasing endothelin-1 production (Lin et al., 2014). Based on these findings, SAH-induced vasospasm may be prevented and treated by achieving a balance between eNOS and iNOS, preventing the suppression of one and the overexpression of the other. SAH induced apoptosis has been found previously to play an important role in a patent’s prognosis. The cell death assay we performed in this study found significant increases in DNA fragmentation, indicating increased apoptosis, in the dentate gyrus in SAH only and SAH plus NC-mimic groups (p < 0.01, compared with controls). SAH induced apoptosis has been found previously to play an important role in a patent’s prognosis. Prunell et al. showed that SAH induced an acute global decrease in cerebral blood flow (CBF) and delayed cell death (Laine et al., 2002; Pentimalli et al., 2004). Apoptosis may not only weaken aneurysm walls (Zhao, Zhang H & Su, 2018) and it can, itself, be involved in the pathogenesis of aneurysms (Zhang et al., 2015). Zhao et al. found marked increases in the proapoptotic genes, caspase-3, -8 and -9, in intracerebral aneurysm mice samples (Zhao, Zhang H & Su, 2018). We found a significant difference in cell death, suggesting that while apoptotic cell death was increased in the dentate gyrus after SAH and miR-195-5p treatment had an anti-apoptotic effect in the dentate gyrus. The previous study demonstrated that miR-195-5p can inhibit apoptosis in damaged neural cells and promote the neural stem cells for neurogenesis (Cheng et al., 2019). It has been also found to attenuate neuronal apoptosis in ischemic stroke rats (Chang et al., 2020) and protect against ischemic stroke clinically Giordano et al. (2019). Therefore, these results indicate that miR-195-5p has therapeutic potential for cell apoptosis after SAH.

Early brain injury, which has been found to contribute to unfavorable outcomes in aneurysmal SAH (Dou et al., 2017; Helbok et al., 2015; Sheng et al., 2018), has previously found to be followed by inflammatory responses that also contribute to progressing injury (Dou et al., 2017; Liu et al., 2019). SAH produces an inflammatory response in the cerebral circulation mediated by proinflammatory cytokines such as TNF-α (Takizawa et al., 2001). TNF-α is known as a pro-inflammatory cytokine involved in neuronal inflammation, apoptosis and necrosis (Zhang et al., 2015). TNF-α expression in the dentate gyrus was significantly higher in this study. Especially, miR-195-5p significantly attenuated the SAH-induced increases in TNF-α in the dentate gyrus. Furthermore, inhibition of TNF-α has been approved for therapeutic use in SAH-induced apoptosis (Lv et al., 2017; Ma et al., 2018; Zhang et al., 2015). Thus, these results indicate that miR-195-5p has therapeutic potential for inflammation after SAH.

MiR-195-5p reversed SAH-induced elevation of p-NF-κB, NF-κB, and p-IκBα in this study. The NF- κB signaling pathway may be the pathway regulated by miR-195-5p. The NF-κB signaling pathway, which has been implicated in passive post-injury necrosis and cell damage and found to activate microglia to secrete inflammatory cytokines and cause amplification of the inflammatory response cascade (Pawlowska et al., 2018; You et al., 2016). The percentage of p-NF-κB and NF-κB protein was found to be significantly increased, but significantly decreased the miR-195-5p group, Additionally, TNF-α as the expression of a proinflammatory cytokine in the dentate gyrus was significantly higher. MiR-195-5p significantly attenuated the SAH-induced increases in TNF-α in the dentate gyrus. Therefore, we believe that it is of paramount importance to develop novel neuroprotective therapies targeting neuroinflammation since this might alleviate early SAH-induced brain injury and cerebral vasospasm-induced delayed cerebral ischemia.

Although MiR-195-5p also has been found to have anti-inflammatory and anti-apoptosis effects in many other contexts, the possibility of using it to treat SAH-induced vasospasm and brain injury had not been studied until our current investigation. The actions mediated by MiR-195-5p are complicated and are involved in a myriad of signaling pathways. It not only regulates cell growth but it also exerts anti-inflammatory and antiapoptosis effects via the NF-κB signal pathway. First, miR-195-5p has been found to suppress tumor by decreasing the expression of multiple NF-κB downstream effectors by directly targeting nuclear factor kappa-B kinase subunit alpha (IKK α) and transforming growth factor-beta(TGF-β)-activated kinase 1 and MAP3K7-binding protein 3 (TAB3)(Ding et al., 2013). Second, it can reduce inflammation by directly blocking the NF-κB pathway, including IKKα and phosphorylated IκBs (Yu et al., 2018). Its ability to block the NF-κB pathway has been harnessed to inhibit the development of abdominal aortic aneurysm (Liang et al., 2017) and chronic pulmonary obstructive disease (COPD) (Li et al., 2020). Third, miR-195-5p has been found to inhibit the expression of pro-apoptotic protein Bax and active caspase-3 through suppressing IKKα/NF-κB pathway and alleviate oxygen-glucose deprivation/reperfusion-induced cell apoptosis (Yang, Cao & Zhang, 2020). More relevant to our study are the findings that miR-195-5p protects against neurovascular injury and apoptosis and has been used to treat neural cells injured by acute stroke, regenerate neurons by promoting neuralstem cell proliferation and migration, reduce inflammation by directly blocking the NF- κB pathway, and improve endothelial functions (Cheng et al., 2019).

Using neuroprotective therapies to treat SAH-induced early brain injury and cerebral vasospasm-induced delayed cerebral ischemia caused by post-SAH neuroinflammation is a relatively new strategy. In this study, we examined the effect of miR-195-5p on SAH-induced vasospasm and brain injury in experimental rats. It was found to reverse cerebral vasospasm and brain apoptosis. It reversed the increases in proteins and alleviated dentate gyrus apoptosis following SAH. By inhibiting the NF-κB pathway, it reduced the level of pro-inflammatory cytokines, including the percentage of p-NF-κB/NF-κB, TNF-α and iNOS, leading to significant reduction in inflammation, vasospasm and apoptosis (Fig. 8).

Figure 8 MiR-195-5p has ameliorated cell death and vasospasm following SAH via the NF-κB signaling pathway.

SAH: Subarachnoid hemorrhage. NF-κB: Nuclear factor kappa-light-chain-enhancer of activated B cells. IκBα: Nuclear factor of kappa light polypeptide gene enhancer in B-cells inhibitor, alpha. NF-κB RE: NF-κB response element. iNOS, Inducible nitric oxide synthase; eNOS, Endothelial nitric oxide synthase; TNF-α, Tumor Neurosis Factor-alpha.

This study has some limitations. One limitation is that it is we did not assess behavior outcomes in our rats. Considering the results we found on vasospasm, there should be clear and convincing improvements in an animal their behavior. Our team members, however, were not totally familiar behaviors testing. Future studies may want to test the results using cognitive and behavioral assessment tools such as the Morris Water Maze and Next Construction tests. Another limitation is that it is we did not examine the concentration of miR-195-5p in the brain. Considering that miR-195-5p mainly acts on vascular endothelial cells, and it is currently unknown whether miR-195-5p can enter the brain through the BBB. Therefore, we did not check the concentration of miR-195-5p in the brain. Future studies may want to test the concentration of miR-195-5p in the brain or blood. In addition, how does miR-195-5p access the central nervous system before being degraded? Considering that miR-195-5p is unstable and peripheral delivery requires carriers to prevent degradation before achieving access to the brain. Intravenous injection is the first chosen route; it can be quickly taken to vascular endothelial cells before miR-195-5p is degraded. However, the permeability of miR-195-5p to the BBB is unknown. We still don’t know if miR-195-5p can enter the brain through BBB. It is one of the limitations of this experiment also.

Conclusion

In this study, miRNA-195-5p attenuated SAH-induced vasospasm by increasing eNOS, decreasing iNOS and inhibiting the NF-κB signaling pathway. It also prevented the neuronal cell death that ensued following SAH by decreasing cytokine TNF-α expression in the dentate gyrus. Thus, we believe that t MiR-195-5p could potentially be used to develop agents to help manage SAH-induced cerebral vasospasm and brain injury, though this would require further research to elucidate in detail the mechanism through it exerts these effects.

Supplemental Information

Supplemental Information 1 Original figures and data.

Click here for additional data file.

Supplemental Information 2 Raw data.

Click here for additional data file.

Supplemental Information 3 ARRIVE 2.0 checklist.

Click here for additional data file.

Abbreviations

CBF Cerebral blood flow

COPD Chronic pulmonary obstructive disease

CV Cerebral vasospasm

CX3CR1 C-X3-C motif chemokine receptor 1

DCI Delayed cerebral ischemia

EBI Early brain injury

ELISA Enzyme-linked immunosorbent assay

eNOS Endothelial nitric oxide synthase

HIF-1α Hypoxia inducible factor 1alpha

IKKα IκB kinase α

iNOS Inducible nitric oxide synthase

MAPK Mitogen-activation protein kinase

MMP-9 Matrix-metalloproteinase-9

NO Nitric oxide

NF-κB Nuclear factor kappa-light-chain-enhancer of activated B cells

IκBα Nuclear factor of kappa light polypeptide gene enhancer in B-cells inhibitor, alpha

p-NF-κB Phosphorylated NF-κB

p-IκBα Phosphorylated IκBα

SAH Subarachnoid hemorrhage

TGF-β Transforming growth factor-beta

TNF-α Tumor Neurosis Factor-alpha

VEGF Vascular endothelial growth factor

Additional Information and Declarations

Competing Interests

Author Contributions

Animal Ethics

Data Availability

The authors declare there are no competing interests.

Tai-Hsin Tsai conceived and designed the experiments, prepared figures and/or tables, and approved the final draft.

Chih-Hui Chang performed the experiments, authored or reviewed drafts of the paper, and approved the final draft.

Szu-Huai Lin performed the experiments, prepared figures and/or tables, and approved the final draft.

Yu-Feng Su analyzed the data, prepared figures and/or tables, authored or reviewed drafts of the paper, and approved the final draft.

Yi-Cheng Tsai performed the experiments, analyzed the data, prepared figures and/or tables, and approved the final draft.

Sheau-Fang Yang analyzed the data, authored or reviewed drafts of the paper, and approved the final draft.

Chih-Lung Lin conceived and designed the experiments, authored or reviewed drafts of the paper, and approved the final draft.

The following information was supplied relating to ethical approvals (i.e., approving body and any reference numbers):

The animals used in this study are approved by the Kaohsiung Medical University Animal Care and Use Committee (IACUC approval No: 100201).

The following information was supplied regarding data availability:

The raw measurements are available in the Supplemental Files.

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
