# Peer review of "Therapeutic effect of and mechanisms underlying the effect of miR-195-5p on subarachnoid hemorrhage-induced vasospasm and brain injury in rats"

_PeerJ, doi:10.7717/peerj.11395_

## Round 0.1 · original submission · Major Revisions

Your manuscript "Therapeutic effect of and mechanisms underlying the effect of miR-195-5p on subarachnoid hemorrhage-induced vasospasm and brain injury in rats” has been reviewed by the two experts. I append their reviews below. I have raised specifics issues about the description of the materials/methods and statistics.
I, therefore, invite you to revise and resubmit your manuscript, considering the points below. Let me reinforce some of the reviewer’s comments and add some more:

1. Please describe how was the sample size calculated?

2. Statistical analyses: the authors should include the values from the appropriate statistical test (e.g., F(x,x) = xx; n = x; P = x.xxx).

3. Considering that small RNAs are unstable and peripheral delivery requires carriers to prevent degradation before achieving access to the brain, how can the authors claim that these compounds are accessing the central nervous system? It seems a limitation of this study.

4. The article should be reviewed by someone fluent in the English language.

Reviewer 1 ·

Basic reporting

The manuscript should be edited by a native speaker of English. Additionally, the raw data of immunobolt was missing.

Experimental design

Methods failed to offer sufficient information to be reproducible by another investigator.

Validity of the findings

The raw data of immunobolt was missing.

Additional comments

In the current study, Tsai et al investigated the role of miR-195-5p in the pathological process following subarachnoid hemorrhage (SAH). And they concluded that administration of miR-195-5p alleviated SAH-induced vasospasm and brain injury. It’s an interesting study. However, several issues must be addressed.
1. Method. How did the author choose the delivey route and dose of miR-195-5p in this study?
2. Method. Had the author determined the concentration of miR-195-5p in brain?
3. Method. Did the authors perfuse rats with formalin before the basilar artery morphometric analysis?
4. Method. How many animals were used for each test, including Basilar Artery Morphometric Analysis, Cell Death Assay, immunoblot, and TNF-α evaluation? They wrote that 96 animals for each test, which was quite complexing.
5. Method. As mentioned in Statistical Analysis, the difference among the groups were analyzed by ANOVA. How were the data analyzed to ensure they were normally distributed, a requirement of ANOVA?
6. Data points should be indicated on all statistical graphs. Providing the bar graph and only statistical parameters can suggest that the data underlying any particular bar are normally distributed and contains no outliers, when this may not be the case.
7. Generally, microRNA selience specific gene through binding to the DNA fragments. How could the miR-195-5p modulate the level of eNOS and iNOS, as well as regulate the phosphorylation p-65 and IκBα?

Reviewer 2 ·

Basic reporting

This study aimed to determine the effects of miR-195-5p on an experimental model of SAH in male rats. It was unclear why authors selected this specific miRNA for studies. No clear rationales or justifications were provided except listing a couple of studies showing beneficial effects of miR-195 mainly in cancers. No prior studies that associate SAH with miR-195 were provided to support the scientific premise of the study.

Experimental design

The miR-195 and control were in suspensions delivered in tail vein injections. It is questionable how this method could ensure BBB penetrability and exhibit efficacy on brain changes as authors assumed. The small RNAs are unstable and peripheral delivery requires carriers to prevent degradation before getting access into the brain. No validation studies were provided about whether miR195 was efficiently getting into the brain and increasing the levels in the brain. With this questionable method for studies, the validity of findings is questionable.

Validity of the findings

see above. The validity of the findings is questionable.

Additional comments

n/a

---

## Round 0.2 · Major Revisions

Dear Authors
I appreciate your efforts in improving the article. However, some aspects still need to be clarified:

a) how was the sample size calculated? What method was used to determine the sample size?

b) the raw data spreadsheet is a little confusing. My impression is that the sample size varies in the same spreadsheet (see for example the spreadsheet Fig. 3 eNOS "). The raw data from Figure 7 is missing. Please review the whole spreadsheet.

---

## Round 0.3 · accepted · Accept

The authors improved the manuscript after the review and can now be accepted.